# Black Fungi on Stone-Built Heritage: Current Knowledge and Future Outlook

**Filomena De Leo *** [ID], **Alessia Marchetta** [ID] **and Clara Urzì** [ID]

Department of Chemical, Biological, Pharmaceutical and Environmental Sciences, University of Messina, Viale F. Stagno d'Alcontres, 31, 98166 Messina, Italy; alessia.marchetta@unime.it (A.M.); urzicl@unime.it (C.U.)
* Correspondence: fdeleo@unime.it

**Featured Application: This is an updated review on black fungi as main biodeteriogens of cultural heritage stone artifacts. Colonization pattern, taxonomy, and methods to eradicate their settlement are discussed here.**

**Abstract:** Black fungi are considered as one of the main group of microorganisms responsible for the biodeterioration of stone cultural heritage artifacts. In this paper, we provide a critical analysis and review of more than 30 years of studies on black fungi isolated from stone-built heritage from 1990 to date. More than 109 papers concerning the fungal biodeterioration activity of stone were analysed. The main findings were a check list of the black fungal taxa involved in the biodeterioration of stone-built heritage, with a particular reference to meristematic black fungi, the main biodeterioration pattern attributed to them, and the methods of study including the new molecular advances. A particular focus was to discuss the current approaches to control black fungi from stone-built heritage and future perspectives. Black fungi are notoriously hard to remove or mitigate, so new methods of study and of control are needed, but it is also important to combine classical methods with new approaches to improve current knowledge to implement future conservation strategies.

**Keywords:** stone cultural heritage; black fungi; MCF; biodeterioration; control

## 1. Background

Stonework, such as artistic sculptures, historical buildings, monuments, archaeological sites, caves, etc., are ubiquitous across the globe, being an expression of culture, religion, aesthetics, and building techniques of populations, typical of certain historical construction periods. Due to their unicity and intrinsic value, ensuring the integrity of stone-built heritage for posterity is a critical issue. The study of biodeterioration of cultural heritage is a hot topic of broad interest to the researcher's community and the implementation of safeguard measures is one of the main goals. All materials are subjected to a natural weathering, and "biodeterioration" of stones should be considered as an integral part of bio-geo-morphogenesis [1–4]. The term "biodeterioration" defines any irreversible transformation of inorganic or organic material with economic, commercial, historic, and artistic loss caused by macro- and micro-organisms [5].

"Biodeterioration" is a very complex matter and conservators should also take into account whether the observed biologically driven phenomena can even be considered positive for the artifact.

In fact, in some cases, the presence of subaerial biofilm (SABs) may have a protective effect on the surface [6]; on the other hand, SABs developed at the interface between rock surface and air is considered the main cause of biodeterioration of stone monuments [7,8].

Microbial biodeterioration of stones is often associated with the presence of a complex community formed by chemoorganotrophic microorganisms (bacteria and microfungi) and autotrophic microorganisms (such as algae and cyanobacteria and to lesser extent

autotrophic bacteria) usually embedded in an extracellular matrix EPS (in which are present DNA, enzymes, pigments, lipids, proteins, etc.). Microbial cells in the EPS show a typical biofilm lifestyle that confers resistance to hostile environments and reinforces the attachment of microorganisms on the surface [6,9].

The prevalence of one or more group of microorganisms depends on numerous factors which include the intrinsic characteristics of the material (such as lithotype, porosity, roughness, and state of preservation) that affect its "bioreceptivity" *sensu* Guillitte [10]. The species composition can vary greatly depending on climatic and microclimatic conditions such as temperature, solar irradiation, shining, nutrient and water availability, and, last but not least, the characteristics of species involved [6]. However, microbial colonization is a very dynamic process in time and space, that is the result of the interactions between microbial species and substrates. It varies continuously during the year following the seasons, and it is also under the influence of the dispersion ability of propagules in the air [11–13].

In recent years, much knowledge has been gained about rock-inhabiting black fungi, and important issues concerning their taxonomy, physiology, phylogeny, and weathering processes [14] have been clarified. However, the majority of studies concerned black fungi from natural environments [4,7,15].

In the field of cultural heritage, most reviews had as a topic the biodeterioration of stone caused by fungi in general [6,16,17]; some have focused on the microbial and fungal deterioration of various type of substrata (both organic and inorganic such as textile, parchment, wood, paper, metals, and stone) used for artworks [18]; few concerned exclusively black fungi as a cause of biodeterioration of stone monuments [19,20].

This paper aims to give an overview on the present knowledge of rock-inhabiting black fungi in the field of stone cultural heritage with reference to their taxonomy, biodeterioration pattern, methods of study, and control, with a look to a future perspectives.

A bibliographic search was carried out using such databases as Scopus (https://www.scopus.com accessed on 17 March 2022), Science Direct (https://www.sciencedirect.com accessed on 17 March 2022), Web of Science (http://www.webofknowledge.com accessed on 17 March 2022), and Google Scholar (https://scholar.google.com accessed on 17 March 2022), that were consulted by using keywords such as 'black fungi', 'meristematic fungi', 'stone monuments', 'stone artworks', 'stone biodeterioration', 'biodeteriogenic fungi', 'fungal treatment', and 'fungal control'.

The search produced about 500 papers of which 109 were included in this paper. The updates of fungal nomenclature were searched in the databases Index Fungorum (http://www.indexfungorum.org accessed on 22 March 2022), National Center for Biotechnology Information (NCBI) (https://www.ncbi.nlm.nih.gov accessed on 24 February 2022) and in the recent literature [14,21].

Nucleotide sequences were retrieved from GenBank database that is accessible from NCBI platforms (http://www.ncbi.nlm.nih.gov accessed on 14 February 2022). Molecular Evolution Genetic Analyses (Mega 11 Software) free downloadable via the URL. http://www.megastoftware.net accessed on 14 February 2022 was employed for alignments and phylogenetic tree constructions.

## 2. Black Fungi and Stone Monuments: An Intimate Connection

Beginning in the 1990s, black fungi were described as one of the most likely groups of microorganisms responsible for the biodeterioration of the stone monuments [22–25] and it was confirmed in the following decades [6,18,26,27].

The term "black fungi" refers to a very huge group of dematiaceous fungi, unrelated phylogenetically, which have in common the presence of melanin in the cell wall that confers an olive brown appearance to the colony [28]. Another common characteristic is the ability to withstand hostile environments such as scarcity of nutrients, high solar irradiation, scarcity of water, high osmolarity, and low pH [15,19,29].

As reported by Gueidan et al. [30] the ancestors of black fungi were well adapted to live in oligotrophic environments such as rock surfaces or sub-surfaces, and currently they can also grow in anthropogenic habitats such as glass, silicon, organic surfaces, metals [31], or consolidants applied on the stone [9].

Their resilience is related to the extremotolerant or even polyextremotolerant characteristics of the species. The stress-tolerance is due to different factors such as: pigmentation, and in particular melanins production; mycosporine-like substances; morphological and metabolic versatility; meristematic development; and oligotrophy [32–34]. All these characteristics make them very suitable for colonizing outdoor rocks and built stones due to the fact that those surfaces can be exposed to extreme environments [17–19].

This group of fungi includes (a) fast growing hyphomycetes of epiphytic origin, recognizable under microscope by the presence of typical conidiophores and spores; (b) pleomorphic hyphomycetes that include the "black yeasts", showing a yeast-like form, and the so-called "black meristematic fungi" with a *Torula*-like growth pattern (Figure 1).

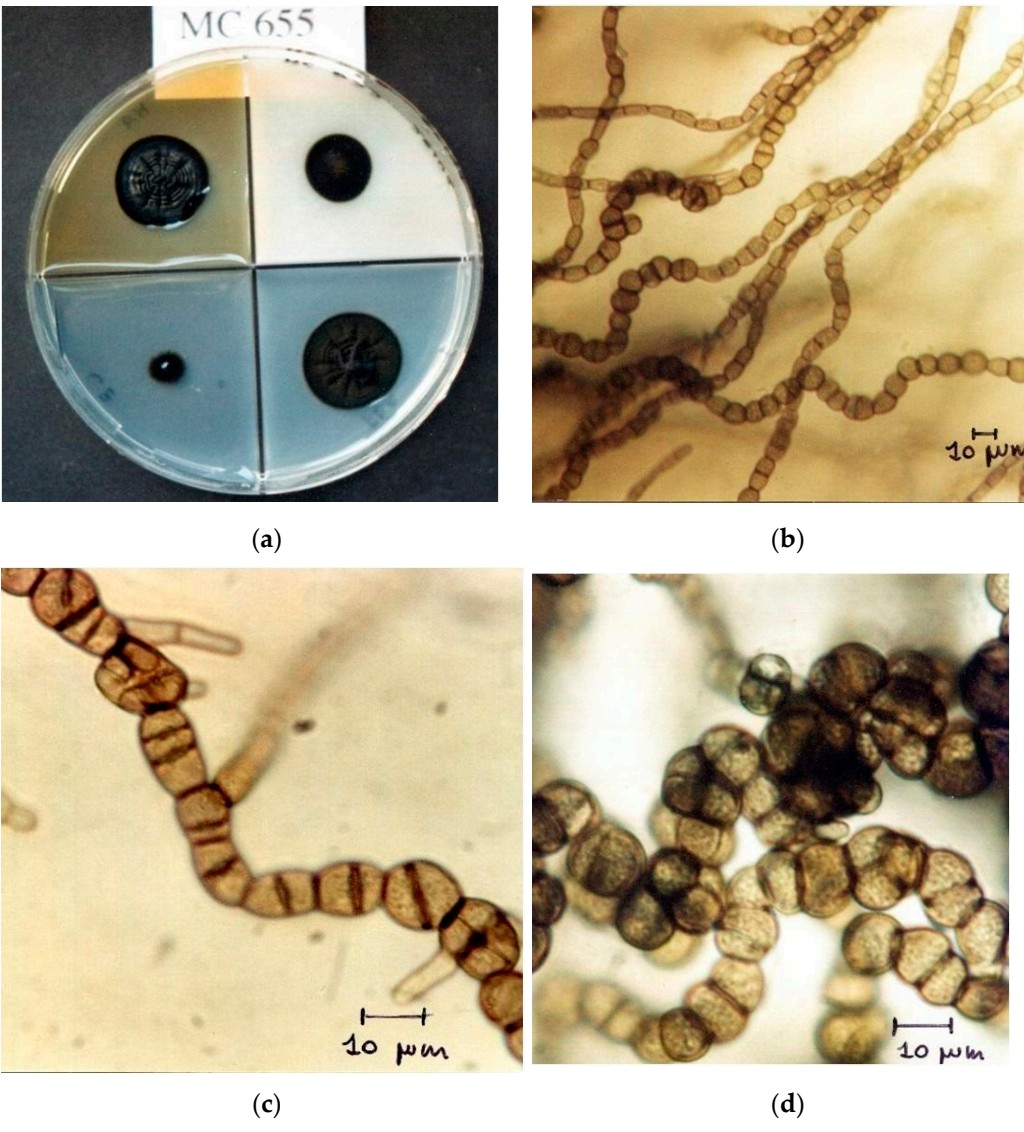

(a)        (b)

(c)        (d)

**Figure 1.** Main morphological characteristic traits of MicroColonial Fungi, MCF. Dark black colonies due to the melanin production as seen (**a**) for the unidentified strain MC 655 on different cultural media after 1 month of incubation. (**b**–**d**) characteristic meristematic pattern of growth described also as *Torula*-like hyphae observed under Light Microscope. Bar is 10 µm.

Hyphomycetes and black yeasts are ubiquitous and widespread all over the world in very different habitats (e.g., soil, fresh water, sea, plants, animals, and humans) [28,35], while the meristematic black fungi, mostly isolated from stone or natural rocks, can be considered the true stone-inhabiting fungi [19,20,36,37].

In the literature, many of black fungi are reported as RIF (rock inhabiting fungi) to emphasize that the "rock" is their preferred or exclusive habitat. However, this terminology does not include their main features such as melanin production, pleomorphism, or meristematic development; for this reason, we do not use it in this context.

In the frame of cultural heritage the acronym MCF (MicroColonial Fungi) as first employed by Staley [38] is widely used for their description. It refers to the typical black cauliform-like colonies visible on the rocks and stones.

Humidity may affect the settlement of MCF on the stone artifacts as unique inhabitants or as associated with other stone colonizers. In fact, in lower or sheltered parts near the ground, where there is a sufficient availability of water, MCF are strictly associated with phototrophic microorganisms with whom, however, they do not establish a symbiotic relationship (Figure 2); in harsh, dry micro-environmental conditions, MCF become the unique colonizers [39–41].

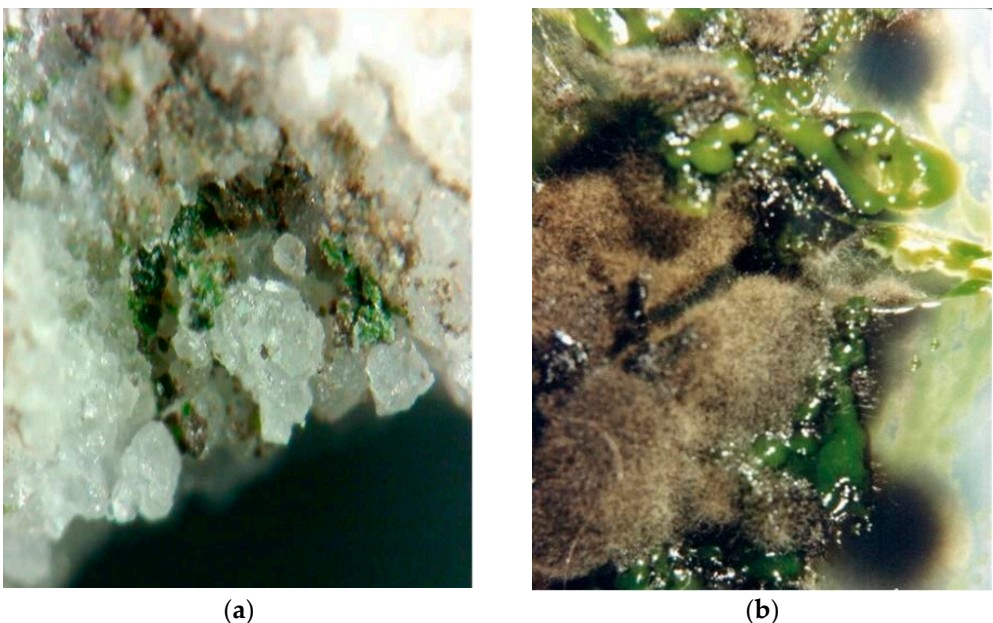

|  (a)  |  (b)  |

**Figure 2.** Close association between MCF and phototrophic microorganisms. (**a**) Association seen directly on a marble sample. Magnification 400X and (**b**) after growth in the isolation medium PDA: *Chlorella*-like alga and *Coniosporium apollinis* MC 728. Magnification 80X.

Black fungi are currently classified in the Phylum of Ascomycota in the Class of Dothideomycetes and Eurotiomycetes, mainly in the order of Capnodiales, Dothideales, Chaetothyriales, Pleosporales and Cladosporiales, and Mycocaliciales [14,20,21,42].

In Table 1 are listed the genera of black fungi identified through molecular analyses that from 1997 up to date have been related to the biodeterioration of stone monuments.

**Table 1.** Genera of black fungi isolated from stone monuments in the period from 1997–2022 in association with visible alterations.

| Class/Order | Genera * | Substrate | Environmental and Climatic Features | Alterations Associated to Fungal Colonization | Refs |
|---|---|---|---|---|---|
| *Dothideomycetes incertae sedis* | *Coniosporium* | Calcarenite, granite, limestone, marble | Mediterranean climate, urban environment | Grayish-black patina, pitting, black spots, greenish to dark green patina, crater shaped lesions, chipping, exfoliation, sugaring, crumbling, superficial deposit, and biofilm | [37,39,43–49] |
| *Dothideomycetes/Capnodiales incertae sedis* | *Capnobotryella* | Limestone, marble | Mediterranean climate, continental climate, and urban environment | Black spots, crater shaped lesions, chipping, exfoliation, sugaring, crumbling, pitting, superficial deposit, and biofilm formation | [45,48,50–52] |
| | *Constantinomyces* | Sandstone | Urban environment, temperate climate | Discolorations, patina | [53] |
| | *Pseudotaeniolina* | Marble, sandstone | Mediterranean climate, arid and desert climate | Biological green patina | [54–56] |
| *Dothideomycetes/Capnodiales* | *Aeminium* | Limestone | Temperate climate | Black discoloration with salt efflorescence | [57] |
| *Dothideomycetes/Cladosporiales* | *Cladosporium* | Calcarenite, granite, limestone, marble, plaster, sandstone, tufa | Ubiquitous worldwide distribution in indoor environments and outdoor | Dark alterations, black spots, black patinas, detachment of marble grains, light grayish patina, crater shaped lesions, chipping, exfoliation, sugaring, crumbling, pitting, superficial deposit, biofilm, black crusts, green biofilm with salt efflorescence, stone erosion and disintegration, and discoloration | [27,40,46,48,49,58–67] |
| | *Verrucocladosporium* | Limestone, marble, sandstone | Mediterranean climate, temperate climate, and urban environment | Black patina, discoloration | [37,53] |

**Table 1.** *Cont.*

| Class/Order | Genera * | Substrate | Environmental and Climatic Features | Alterations Associated to Fungal Colonization | Refs |
|---|---|---|---|---|---|
| *Dothideomycetes/Dothideales* | *Aureobasidium* | Granite, limestone, marble, plaster, sandstone | Urban environment, Mediterranean climate, temperate climate, indoor environment, and urban environment | Black patina, black spots, detachments, superficial deposit, biofilm, discolorations with or without salt efflorescence, black crusts, and stone erosion and disintegration | [37,40,45,49,53,63–65,68] |
| *Dothideomycetes/ Mycosphaerellales* | *Salinomyces* | Marble, sandstone | Mediterranean climate | Black patina | [37] |
| | *Neocatenulostroma* | Limestone, sandstone | Temperate climate, urban environment | Discolorations and/or patina, structural damage | [53] |
| | *Neodevresia* | Limestone, marble, plaster, tufa | Mediterranean climate | Black patina, discolorations, structural damage | [37,53,55,63] |
| | *Saxophila* | Marble | Mediterranean climate | Black patina | [37] |
| | *Vermiconidia* | Limestone, marble, travertine | Mediterranean climate, urban environment | Black patina | [37] |
| *Dothideomycetes/ Neophaeothecales* | *Neophaeotheca* | Marble | Mediterranean climate | Black patina | [37] |
| *Dothideomycetes/Pleosporales* | *Alternaria* | Calcarenite, granite, limestone, marble, plaster, tufa | Ubiquitous worldwide distribution in indoor environments and outdoor | Black spots, black patina, detachment of marble grains, greenish to dark green patina, biofilm, black crusts, green-black patina; and blackish patina | [40,46,49,58–60,63,64,66,67] |
| | *Epicoccum* | Granite, limestone, marble | Urban environment, mediterranean climate, and temperate climate | Black spots, black patinas, detachment, superficial deposit, biofilm, blackish patina, green biofilm, and dark and green biofilm with salt efflorescence | [40,45,49,60,64] |
| | *Phoma* | Calcarenite, granite, limestone, marble, plaster, tufa | Mediterranean climate, temperate climate, urban environment, continental-cold climate, and indoor and outdoor environments | Black spots, black patinas, detachment of marble grains; color changes, crater shaped lesions, chipping and exfoliation, sugaring, crumbling, pitting, superficial deposit, biofilm, and black crusts | [40,46,48,49,58,63] |

**Table 1.** *Cont.*

| Class/Order | Genera * | Substrate | Environmental and Climatic Features | Alterations Associated to Fungal Colonization | Refs |
|---|---|---|---|---|---|
| *Dothideomycetes/Venturiales* | *Ochroconis* | Calcarenite | Subterranean environment | Black patina | [69] |
| *Eurotiomycetes incertae sedis* | *Sarcinomyces* | Marble | Mediterranean climate | Black spots | [70] |
| *Eurotiomycetes/Chaetothyriales* | *Cyphellophora* sp. | Plaster | Mediterranean climate | Black/grayish patina | [63] |
| | *Exophiala* | Calcarenite, limestone, marble, sandstone | Mediterranean climate, urban environment, temperate climate, and hypogean environment | Dark alterations, black spots, black patinas, detachment of marble grains, discolorations, and visible structural damage | [27,37,40,45,53,71] |
| | *Lithophila* | Limestone, marble | Mediterranean climate, urban environment, and dry continental climate | Black spots, black patinas, detachment of marble grains | [37,40,72] |
| | *Knufia* | Limestone, marble, sandstone travertine | Mediterranean climate, urban environment, continental temperate climate, and dry continental climate | Black and grey spots, dark macropitting, biopitting, crater shaped lesions, chipping, exfoliation, sugaring, crumbling, discolorations, patina, and visible structural damage | [37,41,43,45,48,53,72–74] |
| | *Rhinocladiella* | Marble | Mediterranean climate | Black spots, crater shaped lesions, chipping and exfoliation, sugaring, crumbling, and pitting | [48] |
| *Eurotiomycetes/Mycocaliciales* | *Mycocalicium* | Marble | Mediterranean climate, urban environment | Black spots, crater shaped lesions, chipping and exfoliation, sugaring, crumbling, and pitting | [45,48] |

* According to the current taxonomic nomenclature.

In manuscripts published prior to 1999, black meristematic fungal species that were identified without molecular analyses, such as *Hormonema dematioides*, *Lichenothelia* sp. and *Hortaea werneckii*, *Trimmatostroma* sp., are listed among the most abundant fungal species present in arid and semiarid environments in association with biodeterioration of stone monuments [3].

The molecular analyses introduced at the end of the 20th century considerably increased the knowledge about the taxonomy of the black fungi isolated from stone monuments and allowed the description of twenty-six new species and three new genera.

The new species and genera described are listed below:

*Sarcinomyces petricola* Wollenzien and de Hoog [73]; *S. sideticae* Sert and Sterflinger [70]; *Coniosporium apollinis* Sterflinger, *C. perforans* Sterflinger [43]; *C. uncinatum* De Leo, Urzì and de Hoog [44]; *C. sumbulii* Sert and Sterflinger [47]; *Phaeococcomyces chersonesos* Bogomolova and Minter [74]; *Pseudotaeniolina globosa* De Leo, Urzì and de Hoog [54]; *Capnobotryella antaliensis* Sert and Sterflinger [50]; *C. erdogani* Sert and Sterflinger; *C. kiziroglui* Sert and Sterflinger [51]; *Ochroconis lascauxensis* Nováková and Martin-Sanchez; *O. anomala* Nováková and Martin-Sanchez [69]; *Knufia marmoricola* Onofri and Zucconi, *K. vaticanii* Zucconi and Onofri; *K. karalitana* Isola and Onofri; *K. mediterranea* Selbmann and Zucconi [37]; *K. calcarecola* Su, Sun and Xiang [72]; *Exophiala bonarie* Isola and Zucconi; *Vermiconia calcicola* de Hoog and Onofri [37]; *Devriesia simplex* Selbmann and Zucconi; *D. modesta* Isola and Zucconi [55]; and *D. sardiniae* Isola and de Hoog [37].

Three new genera and 4 species were also introduced as new: *Saxophila tyrrenica* Selbmann and de Hoog, *Lithophila guttulata* Selbmann and Isola [37], *L. catenulata* Su, Sun and Xiang [72], and *Aeminium ludgeri* Trovão, Tiago and Portugal [57].

Over the years, some of the above mentioned genera and species were reclassified: in particular, *Sarcinomyces petricola* and *Phaecoccomyces chersonesos* resulted identical, and they were reclassified as *Knufia petricola* [75,76]; *Coniosporium perforans* is now a synonym with *Knufia perforans* [76]; *Devriesia* species and *Vermiconia* species were included, respectively, in the new genera of *Neodevriesia* [77] and *Vermiconidia* [21]. Hao et al. [78] proposed a revision of the genus *Ochroconis* that was established as synonymous with the sister genus of *Scolecobasidium.* However, this taxonomic accommodation has been refused by Samerpitak et al. [79,80] on the basis of phylogenetic analyses and because the old generic name *Scolecobasidium* is considered of doubtful identity for the ambiguity of type specimens; therefore, the genus *Ochroconis* that is also characterized by oligotrophism and mesophilia was maintained.

However, many questions regarding the taxonomy and phylogeny of black fungi are still unresolved and further studies are required, especially to clarify the taxonomical position and phylogeny of many species of *incertae sedis* and of strains that are preserved in the mycological collections and are not yet identified (Figure 3, Table 2).

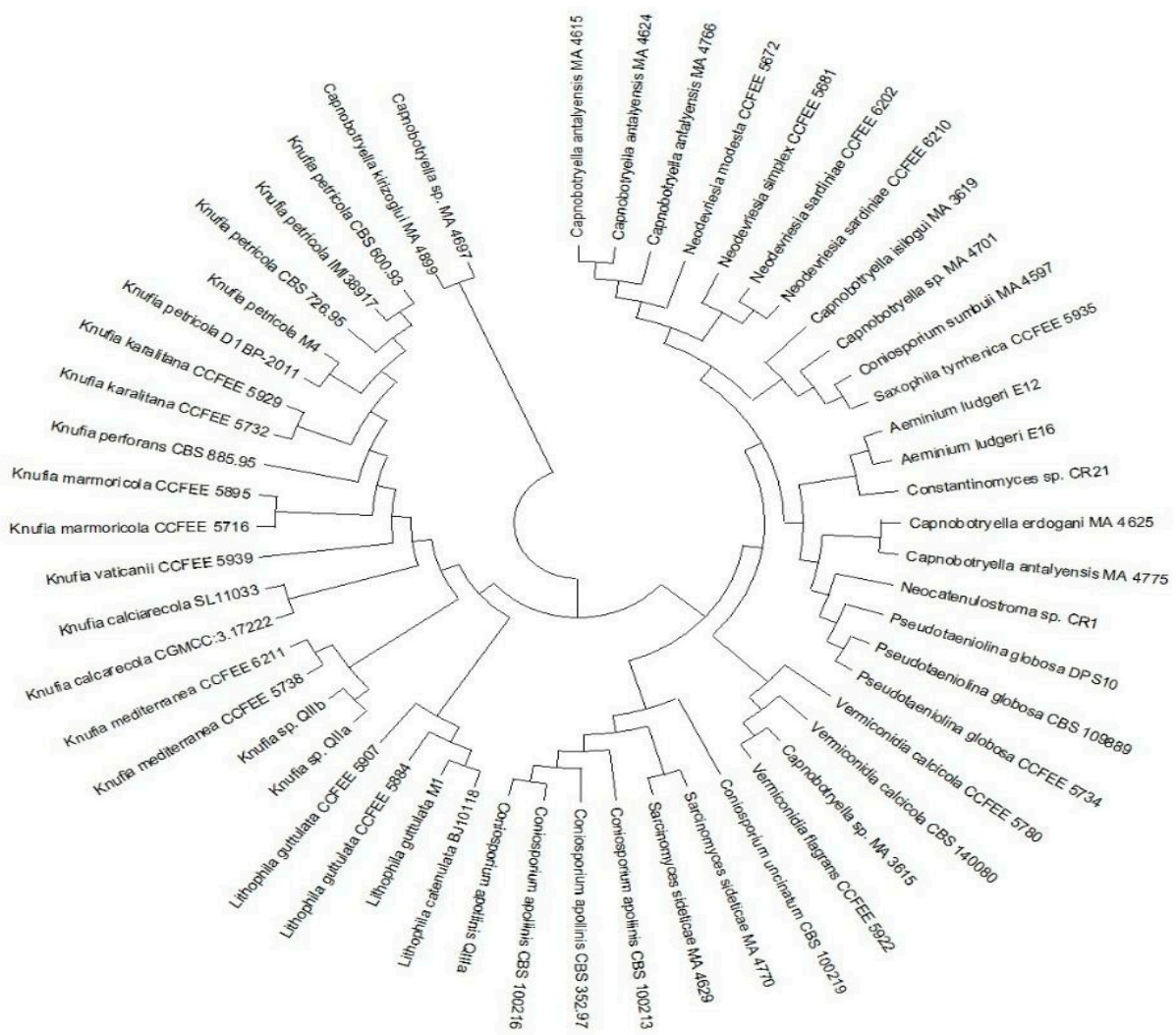

**Figure 3.** Phylogenetic tree (Neighbour-joining, Kimura two-parameters) showing the genetic divergence among ITS rDNA sequences of meristematic black fungi retrieved from GenBank database (https://www.ncbi.nlm.nih.gov/nucleotide accessed on 14 February 2022) and listed in Table 2.

**Table 2.** ITS rDNA sequences of representative MCF isolated from stone monuments aligned in Figure 3.

| Taxon | Strain | ITS rDNA |
|---|---|---|
| *Capnobotryella antalyensis* | MA 4615 | AJ972858 |
| *Capnobotryella antalyensis* | MA 4624 | AJ972850 |
| *Capnobotryella antalyensis* | MA 4766 | AJ972851 |
| *Capnobotryella antalyensis* | MA 4775 | AJ972860 |
| *Capnobotryella isilogui* | MA 3619 | AM746201 |
| *Capnobotryella erdogani* | MA 4625 | AJ972857 |
| *Capnobotryella kirizoglui* | MA 4899 | AJ972859 |
| *Capnobotryella* sp. | MA 4701 | AJ972856 |
| *Capnobotryella* sp. | MA 4697 | AJ972855 |
| *Capnobotryella* sp. | MA 3615 | AM746203 |
| *Neodevriesia modesta* | CCFEE 5672 | KF309984 |
| *Neodevriesia simplex* | CCFEE 5681 | KF309985 |
| *Neodevriesia sardiniae* | CCFEE 6202 | KP791765 |
| *Neodevriesia sardiniae* | CCFEE 6210 | KP791766 |
| *Saxophila tyrrhenica* | CCFEE 5935 | KP791764 |

**Table 2.** *Cont.*

| Taxon | Strain | ITS rDNA |
|---|---|---|
| *Aeminium ludgeri* | E12 | MG938054 |
| *Aeminium ludgeri* | E16 | MG938061 |
| *Neocatenulostroma* sp. | CR1 | KY111907 |
| *Constantinomyces* sp. | CR21 | KY111911 |
| *Pseudaeniolina globosa* | DPS10 | MH396690 |
| *Pseudotaeniolina globosa* | CBS109889 | NR136960 |
| *Pseudotaeniolina globosa* | CCFEE5734 | KF309976 |
| *Vermiconidia calcicola* | CBS 140080 | NR_145012 |
| *Vermiconidia calcicola* | CCFEE 5780 | KP791761 |
| *Vermiconidia flagrans* | CCFEE 5922 | KP791753 |
| *Coniosporium uncinatum* | CBS 100219 | AJ244270 |
| *Coniosporium apollinis* | CBS 100213 | AJ244271 |
| *Coniosporium apollinis* | CBS 352.97 | NR159787 |
| *Coniosporium apollinis* | CBS 100216 | AJ244272 |
| *Coniosporium apollinis* | QIIIa | MH023395 |
| *Lithophila catenulata* | BJ10118 | JN650519 |
| *Lithophila guttulata* | M1 | MW361305 |
| *Lithophila guttulata* | CCFEE 5884 | KP791768 |
| *Lithophila guttulata* | CCFEE 5907 | KP791773 |
| *Knufia mediterranea* | CCFEE 5738 | KP791791 |
| *Knufia mediterranea* | CCFEE 6211 | KP791793 |
| *Knufia vaticanii* | CCFEE 5939 | KP791780 |
| *Knufia calcarecola* | SL11033 | JQ354925 |
| *Knufia calcarecola* | CGMCC 3.17222 | KP174862 |
| *Knufia marmoricola* | CCFEE 5895 | KP791775 |
| *Knufia marmoricola* | CCFEE 5716 | KP791786 |
| *Knufia perforans* | CBS 885.95 | AJ244230 |
| *Knufia karalitana* | CCFEE 5732 | KP791782 |
| *Knufia karalitana* | CCFEE 5929 | KP791783 |
| *Knufia petricola* | CCFEE 726.95 | KC978746 |
| *Knufia petricola* | CBS 600.93 | KC978744 |
| *Knufia petricola* | IMI38917 | AJ507323 |
| *Knufia petricola* | D1 | JF749183 |
| *Knufia petricola* | M4 | FJ556910 |
| *Knufia* sp. | QIIa | MH023393 |
| *Knufia* sp. | QIIb | MH023394 |

## 3. Mechanisms Involved in the Stone Biodeterioration

Being well adapted to the stone habitat and being oligotrophic, this group of fungi can often act as pioneer colonizer of the stone. In fact, for their growth it is sufficient to have just a little input of nutrient coming from the surrounding environment (e.g., animal and plant particles, air pollutants, guano droppings, etc.) [81,82]. Marble exposed to different environments and laboratory experiments demonstrated that black fungi such as *Aureobasidium pullulans* can be the first colonizer of freshly exposed marble surfaces in outdoor conditions [68,83].

The presence of a source of organic matter, such as the proximity of plants and trees, can considerably increase the chances of colonization by these fungi and the consequent rate of biodeterioration (Figure 4).

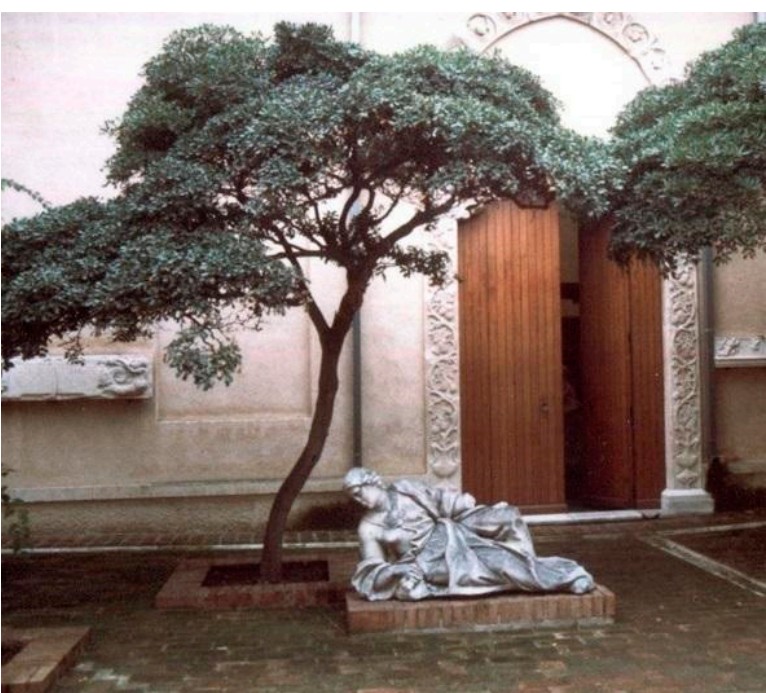

**Figure 4.** Extended black-greyish patina due to black fungi colonization on marble statue located in the inner yard of the Museum of Messina, Italy, under a tree of *Pittosporum tobira.* Fungal strains isolated from the statue were attributed to meristematic black fungi and genera of *Cladosporium*, *Alternaria* and *Phoma* [81].

The mechanism of biodeterioration of stone monuments caused by black fungi is not fully understood [16,84].

The pattern of colonization of black fungi, and in particular of MCF, demonstrate once more that these microorganisms are well adapted to the stone habitat. In fact, they not only grow on the surface of stone, causing an aesthetic alteration due to the presence of melanin in the mycelium and conidia, described as discoloration, black staining, black spots, and black/greyish patinas [9,20,85]; but they can also act as true endolithic microorganisms by penetrating into the rocks/stones via intercrystalline spaces or through an active mechanism in which both mechanical and chemical aspects are hypothesized.

Microscopic observations show that where they settle is shaped accordingly to the morphology of these fungi (Figure 5). This fact can be explained by a local release of organic acids, followed by a precipitation of mineral phases; this buffer effect may be the reason why these fungi, in contrast to other ubiquitous hyphomycetes, such as *Aspergillus niger* and *Penicillium* spp., as reported by Salvadori and Municchia [16], do not show a marked organic acid production in laboratory conditions. However, Favero Longo et al. [86] demonstrated that some species of MCF (e.g., *Knufia petricola*) penetrate actively into freshly exposed stone probes through the production of iron-chelating molecules (siderophores such as compounds).

Another mechanism is due to the ability of these fungi to penetrate the stone using already existing fractures and cracks. The mechanical forces due to the expansion of hyphae may increase the fractures and cause the loss of materials [18,26,39]. To explain the mechanical penetration of the hyphae into the stone, a past hypothesis gave a crucial role to melanin present in the cell wall of hyphae and in the meristematic cells, similar to the process of penetration of phytopathogenic fungi in the host cells [16,23,25,39].

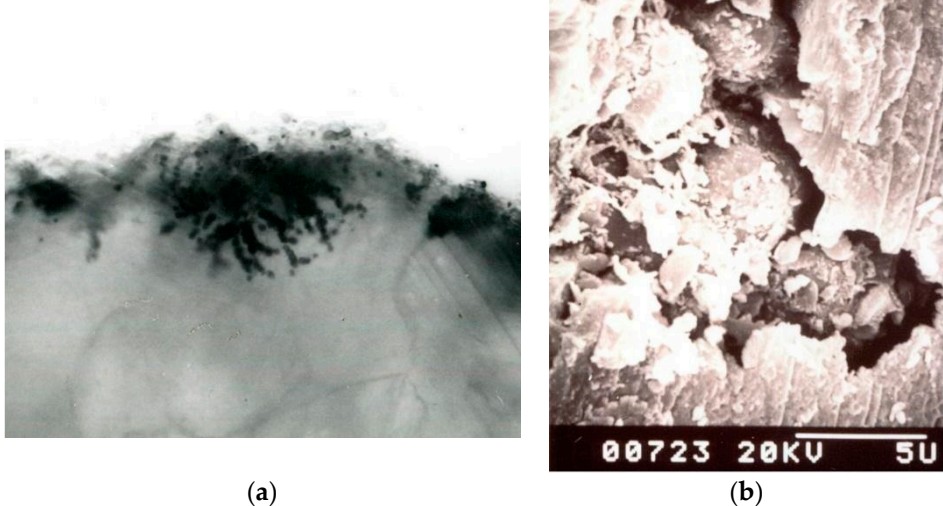

(**a**)               (**b**)

**Figure 5.** Microscopic observation of the endolithic behaviour of meristematic black fungi. (**a**) Settled on a thin section of Carrara marble sarcophagus flake showing the meristematic chains deepening inside the marble, magnification 400X; (**b**) SEM image showing MCF into the marble with dissolution pattern following the same shape of meristematic cells.

In fact, the melanin present in the appressoria of the phytopathogenic fungi confers turgor and rigidity to the cells, favouring their penetration of the host cells [87].

However, on the basis of the results obtained in a recent publication by Tonon et al. [88], this hypothesis should be rejected. In fact, non-melanized mutants of the black fungus *Knufia petricola* do not lose their ability to penetrate carbonate pellets regardless of porosity; on the contrary, thin, non-melanized, exploring hyphae showed an even higher penetration pattern into the stone, probably due to their nutrient-seeking role and thinness.

## 4. Multistep Analyses to Study Black Fungi from Stone Monuments

Contrary to studies carried out on natural rocks, in the frame of cultural heritage it is mandatory to carry out all the necessary multidisciplinary studies (chemical, geological, physical, biological analysis, etc.), by using very low destructive or non-destructive sampling methods [18,46,67,89]. This fact may limit the extent of the studies, but a careful planning of a sampling campaign leads to the right protocol of intervention and of assessing the risk of further biodeterioration processes.

A general useful multistep approach should include microscopy, cultural analyses, molecular analyses, and the laboratory evaluation of selected methods of control (chemical or physical) on isolated strains. The assessment of environmental conditions (temperature, humidity, shining, presence of surrounding vegetation, or other organic sources, atmospheric pollutants, etc.) should be also evaluated. In fact, these data not only allow us a better understanding of the physiology and ecology of fungi, but can help to control their growth indirectly, especially in indoor and/or in confined environments. At the end, a monitoring campaign over time should establish the level of the risk of the item and the frequency of intervention.

Further, a common language for the description of the alterations is also indispensable for sharing the results with the scientific community; to this purpose, there is a glossary [90,91] to obtain an objective and standardized description.

### 4.1. Methods of Isolation and Characterization

The sampling is critical. As reported in the previous paragraph, due to the value of the artifacts, non-invasive sampling methods have been developed over the years and are now widely used. Examples are a needle to take samples from black spots or cavities, scalpel or lancet to scrape fungi from the surface, and adhesive tape sample, useful both

for microscopy and cultural and molecular analysis as it provides a mirror image of the stone colonization [22,92,93].

Microscopic examination of the samples is the most common practice to obtain evidence of black fungi directly from stone samples. In fact, due to their size, morphology, and pigmentation, black fungi can be directly visualized under optical microscope without a specific preparation. Even with a good light microscope (LM) or, better, with a scanning electron microscope (SEM), detailed information is obtained. Microscopy is useful for the direct visualization of the fungi in the stone sample, for determining what types of relationship they establish with stone material and with other types of microorganisms, and also for addressing the next step of analyses. Unfortunately, fluorescence microscopy (FM) cannot help in detecting and studying black fungi due to the presence of melanin that masks the fluorochrome fluorescence.

Still, cultural analyses remain the best way to study this group of fungi.

Black fungi, and in particular MCF, possess a poor ability to compete with fast growing fungi, being characterized by a slow growth rate, and very often require more than 1 month of incubation before visible colonies are seen. These reasons explain why they are difficult to isolate and maintain in culture [94]. Nevertheless, selective cultural media that inhibit the growth of bacteria and of fast-growing fungi are successfully employed both for qualitative and quantitative cultural analyses [22,95].

These culture techniques have the advantage of allowing the whole characterization of the isolates by microscopical, biochemical, physiological, and molecular analysis; these latter are indispensable for the identification of MCF that do not have recognizable morphological traits. Multilocus sequencing typing (mlst) is routinely carried out to resolve their taxonomic and phylogenetic position [55,56] by Blast search homology (Basic Local Alignment Search Tools) available online. However, many nucleotide sequences in the Genkank nucleotide database are not updated in the "definition". Therefore, although this approach is within the reach of all laboratories, it requires a deep knowledge of the literature and a curated nucleotide database.

A deeper genetic characterization of the isolates can be obtained by whole genome analyses but, to date, only four genomes belonging to three species of black fungi (*Coniosporium apollinis* CBS 100218, *Knufia petricola* MA5789, *K. petricola* MA5790, and *Aeminium ludgeri* DSM 106916) isolated from stone monuments are available in the NCBI database (https://www.ncbi.nlm.nih.gov/genome/ accessed on 15 February 2022).

Cultural techniques also allow us to investigate the biodeteriorative abilities of the isolated species by setting up laboratory experiments that simulate their settlement, colonization, and biodeterioration pattern [96,97].

### 4.2. Culture Independent Analyses

It is well known that only a very small percentage (<0.5%) of environmental microorganisms can be cultured. Therefore, culture independent molecular approaches have been developed to overcome this limitation; the metagenomic approaches allow the study of the microbiota (microbial community) and its microbiome (gene pool) by total acid nucleic extraction from the samples, both DNA and/or RNA, depending on the purpose.

Recently Sterflinger and Pinar [98] reviewed the main molecular-based techniques that are currently available and, although some of these have not yet been used in the field of cultural heritage, they could be adapted for future studies.

In the literature, the majority of papers are focused on Bacteria and Archaea, while there are very few papers about microfungi from stone monuments [99,100].

Large-scale genomic analyses, such as high-throughput-sequencing analyses, are becoming increasingly popular to study the microbiota (both for biodiversity and functional genes analyses) in different areas of research, and cultural heritage is not an exception [89,99]. These techniques were developed in the late 1990s and early 2000s, and today they are more accessible to many laboratories due to the lower costs and the facilities offered by

many companies that can carry out all steps of analysis, from nucleic acid extraction to bioinformatic data analysis.

The culture-independent approach certainly contributes to the deep knowledge on the microorganisms associated with the biodeterioration processes. However, there is a very high risk to obtain a plethora of data that are not easily interpretable, because they cannot be directly connected with the biodeterioration phenomenon observed. In fact, it is obvious, but not trivial, that the discovery of a microbial agent on a monument could be not related to any biodeteriogenic activity. Therefore, the current need is to associate these techniques to the culture-based ones for a more complete characterization of the state of deterioration of the artifact and for implementation of the more suitable prevention strategies.

## 5. How to Control Black Fungi

Despite the wide literature regarding the control of biodeteriogens on inorganic surfaces as reported in recent books and reviews [6,9,101–103], very little is said regarding the effectiveness of treatments against black fungi.

Black fungi, especially meristematic ones, are very difficult to eradicate and tend to be one of the first colonizers after cleaning procedures [6,12,83]. In the Lascaux cave, a black yeast *Ochroconis lascauxensis* caused an important and extensive black discoloration on the cave's walls whose origin and evolution were probably linked to the intensive biocide treatments [104].

In order to achieve protection of an artifact, both indirect and direct methods should be implemented. The first ones aim to control, or more realistically to mitigate, the fungal growth by modification of the chemical-physical parameters such as humidity, source of nutrients, and temperature, that are crucial key factors for fungal growth. It is obvious that this is rarely fully achievable, and only in particular circumstances, such as in indoor environments (churches, museums, etc.) or for movable artifacts and objects that can be moved if necessary, while in outdoor conditions this is quite impossible.

Direct treatments aiming to kill/reduce black fungi on the stone should be different on the basis of their colonization pattern (diffuse patina, spot-like colonization, or intercrystalline growth) and on the characteristics of the environment; for example, in an indoor environment, the air is often heavily contaminated by fungal spores and thus they need to be eliminated at the same time as those settled on the surfaces; otherwise, their presence in the air is a continued source of reinfection.

Among the potential methods commonly used to control biodeterioration, physical methods such as mechanical removal and UV and heat shock treatments [101,105], are not very effective against black fungi [102,106].

Regarding chemical methods, in laboratory conditions, classical biocides (e.g., Preventol RI 50, Biotin R, Rocima™ 103) are still the most effective [102,107] and in the field they produce efficient results during cleaning procedures. Plant based extracts show a scarce effectiveness against fungi, and this difficult group of microorganisms is not even taken into account to assess their activity [108]. Nanoparticles are commonly used as biocides due to their activity against algae, cyanobacteria, and most bacteria, but they are not really satisfactory against black fungi.

Protective coatings with antifouling properties may have various effects. In fact, $TiO_2$ based coatings, pure or doped with Ag, show a good effect but are limited to a short/medium term after application [108]. However, in both laboratory and field conditions, after treatments with titania-based coatings, black fungi are the first to recolonize the stone surface in dry environments, while algae first appears in damping walls [63]. Very recently, in laboratory conditions, cholinium@Il based coatings have shown that the use of Il's with a 12 C chains and DBS as anion in combination with nanosilica coatings (e.g., Nano Estel) could be effective against the colonization of black fungi for a period of time over 30 months [109].

One possible explanation of this scarce effectiveness of most treatments against black fungi is that they possess a genetic resistance to environmental stresses, as reported in

the previous paragraphs. Therefore, the different mechanisms concurring to the stress protection response may interfere to the biocidal treatments.

Understanding the cause of their resilience could improve the strategies for their control.

### 6. Concluding Remarks

The study of biodeterioration of stone monuments is quite complex and cannot be improvised. For a correct understanding of biodeterioration phenomena and the implementation of measures aimed at the elimination and/or mitigation of biodeteriogenic microorganisms, it is important to consider the monument and its surrounding as a whole.

When working for the protection of cultural heritage artifacts, scientists should not follow the same protocols for all the situations. In general, it is necessary to:

(1)   Listen the conservators;
(2)   Evaluate the environmental climatic conditions and specific conditions, such as the type of material and the overall status of conservation of monument; the description of the type of alteration visible under naked eye should be also included;
(3)   Interact with the other experts involved;
(4)   Answer the questions posed by the conservators.

The analysis must be planned according to their questions.

It is also important to relate the presence of fungal species with the observed biodeterioration phenomenon; then, for treatments, it is possible to use well known protocols or propose new products/treatments that, however, need to be tested in laboratory with fungal isolates and in situ (on probes, not on the item!!!) before applying it to the CH item. Finally, consider evaluating the use of coatings that match Green conservation criteria and are effective to prevent or slow down new colonization.

As there is not only one method that is valid in all circumstances, we have to work out, case by case, the best solution and monitor the result over the time to avoid unexpected and/or undesirable effects as much as possible.

Black fungi, especially meristematic ones, are very dangerous for stone artifacts for several reasons:

(a)   They are responsible of discolouring of the stone surface. The extended colonization of surfaces changes the global vision of the artifact, especially if different material and colour of stones were used by the artist;
(b)   Moreover, black fungi show an inter-crystalline pattern of growth. This pattern causes crystals to detach (so called sugaring) with loss of precious material, especially because it involves the first surface layer (very important for bas-reliefs and sculptures);
(c)   They could determine the biopitting. Fungi excavate cavities on the stone where they can better settle, giving the surface a pockmarked aspect. The convergence of several biopitting can often lead to larger cavities;
(d)   Hyphae penetrate deep into the surface, even more than a few mm;
(e)   Chemical and physical treatments used for other microorganisms are often non efficient in eradication;
(f)   Black fungi are often the first colonizers after the treatments.

For all these reasons, new methods of study and of control are needed that also aim to search for more eco-friendly molecules and/or approaches.

Despite the increase in interest in black fungi as a cause of biodeterioration of stone monuments and artifacts, many aspects need a more in-depth analysis. For example, not much is known about the molecular mechanisms involved in stress tolerance, in colonization, and biodeterioration of stone. Very important results were achieved by laboratory experiments that, however, concern only some species, and are still too few to generalize the results obtained. Only four genomes of three species were sequenced, and they are not sufficient for a comparative analyses aiming to a better understanding of the above mentioned processes and mechanisms.

Furthermore, the creation of a curated database including the nucleotide sequences used for identification of black fungi from monuments could be very helpful, as the public databases are not curated and are outdated; they could even be misleading for people who are not interested in deepening the knowledge around the taxonomy and phylogeny of this group of fungi.

Finally, it is also important to combine the classical methods with the new ones to improve current knowledge useful for implementation of future conservation strategies.

**Author Contributions:** Conceptualization, F.D.L. and C.U.; methodology, F.D.L., A.M. and C.U.; resources, F.D.L. and C.U.; data curation F.D.L.; writing—original draft preparation, F.D.L., A.M. and C.U. writing—review and editing, F.D.L., A.M. and C.U. All authors have read and agreed to the published version of the manuscript.

**Funding:** No funding was used for this manuscript.

**Informed Consent Statement:** Not applicable.

**Data Availability Statement:** Not applicable.

**Acknowledgments:** Authors would like to thank Sherron Collins for her revision of the English text.

**Conflicts of Interest:** The authors declare no conflict of interest.

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
