# Peer review of "Black Fungi on Stone-Built Heritage: Current Knowledge and Future Outlook"

_applsci, doi:10.3390/app12083969_

Round 1
Reviewer 1 Report
The article entitled: “Black Fungi on Stone-Built Heritage: Current Knowledge and Future Outlook” is in line with the Applied Sciences journal. It is based on a literature review. The topic undertaken by the authors is important from a practical point of view, especially for the protection of the heritage. The article is quite well planned, but the article requires minor changes before publication:
- Abstract: is not enough informative, the research methods should be clarified (literature review, critical analysis, etc.), and the main findings need to be supplemented;
- Introduction – Please add some information about the justification for this topic of interview (importance);
- Research methods should be clarified (literature review, critical analysis, etc.), also used database and keywords for literature selection;
- Figures 1, 2, 5 – own research or external source? If external source the reference is needed;
- Figure 3 – please enlarge;
- Figure 4 – all sculpture is presented, but it will be nice to have second figure where the fungi attack will be visible;
- Could you propose some procedure to protect the artifacts?
- Please add some short assessment how dangerous is black fungi comparison to other form of deterioration the artifacts.
- Conclusions - What kind of further research is needed?
Author Response
Thank you and reviewers for your pertinent questions, suggestions and corrections. We have responded to all the questions raised in the following text and we have highlighted in the manuscript all corrections done in blue ( text) and yellow (references).
Some parts have been expanded, others have been re-written, others are new.
In addition revision of the English text was done by a native English speaker.
Detailed answers are in the attached file.
Answers to Reviewer 1
Comments and Suggestions for Authors
The article entitled: “Black Fungi on Stone-Built Heritage: Current Knowledge and Future Outlook” is in line with the Applied Sciences journal. It is based on a literature review. The topic undertaken by the authors is important from a practical point of view, especially for the protection of the heritage. The article is quite well planned, but the article requires minor changes before publication:
-Abstract: is not enough informative, the research methods should be clarified (literature review, critical analysis, etc.), and the main findings need to be supplemented;
- The abstract was re-written according to your suggestions.
- Introduction – Please add some information about the justification for this topic of interview (importance);
- Thank you for this suggestion. The justification of the topic was added in the introduction: Lines 66-79.
-Research methods should be clarified (literature review, critical analysis, etc.), also used database and keywords for literature selection;
- The methodology used for bibliographic research, taxonomy and nucleotide sequences analyses were added in the Lines 80-95.
- Figures 1, 2, 5 – own research or external source? If external source the reference is needed;
- The pictures come from our research activities.
- Figure 3 – please enlarge;
- Done
- Figure 4 – all sculpture is presented, but it will be nice to have second figure where the fungi attack will be visible;
- Sorry, we have not a close up picture of this statue.
- Could you propose some procedure to protect the artifacts?
- Some sections were added to clarify this issue: Lines 422-429; Lines 473-490.
- Please add some short assessment how dangerous is black fungi comparison to other form of deterioration the artifacts.-
- In the Conclusion we added this part: Lines 494-509.
- Conclusions - What kind of further research is needed?
- Thank you for your question. In the Conclusion we discussed this issue: Lines 512-528.

Reviewer 2 Report
The aim of the present work is to review the current knowledge on black fungi as biodeteriogens of stone cultural heritage artifacts. Although an overview of the current knowledge is worthy, previous reviews should be taken into account and this work should reviewed the knowledge not addressed in the previous one. Thus, the starting point could be knowledge from 2000 and not reviewed in articles 3, 18 and 41.
Moreover, there are sections not fully addressed. That is the case, for instance, of the Mechanism involved in the stone deterioration section. Some mechanisms are pointed but not developed. For instance, last paragraph in this section:
Another mechanism is due to the ability of these fungi to penetrate into the stone by using already existing fractures, cracks. The mechanical forces due to the expansion of hyphae may increase the fractures and cause the loss of materials [18, 19, 36]. The melanization of the cell walls and the meristematic development explain in part such mechanisms [17,85,86].
Authors should give information about how and why melanization and meristematic development explain that mechanism.
There are many sentences pointing to very important issues but they are not developed. For instance: (second paragraph in section 4):
The most useful approach is a multistep analysis among which the assessment of the surrounding environmental conditions must be evaluated [45,87].
Which environmental conditions?, why?, how?....
Other paragraph as the first one in the page 7, has not verb; it is a list of species but I don’t know what the authors wanted to say.
In relation to table 1, the most frequent species are shown. The most frequent?, most of them were just cited in one article. Which criterium was followed to define one specie as frequent?.
Because of the above I consider that the present review can not be accepted in the present form and only can be reconsidered after major revision, especially in terms of focusing the review on subjects not previously reviewed.
Author Response
Thank you for your pertinent questions, suggestions and corrections. We have responded to all the questions raised in the following text and we have highlighted in the manuscript all corrections done in blue ( text) and yellow (references).
Some parts have been expanded, others have been re-written, others are new.
In addition revision of the English text was done by a native English speaker.
Detailed answers to your comments are listed in the attached file

Round 2
Reviewer 2 Report
This is a really improved version of the manuscript which I think should be published just after minor revisions:
- Line 15: 'More than 500 papers concerning the fungal biodeterioration activity of stone were analyzed' . 500 should be changed by 107 as authors said in line 80: 'The search produced about 500 papers of which 107 were included in this paper'.
- Line 90: 2090s????
- Line 168: 'Three new genus...' should be changed by Three new genera.
Author Response
Answers to the Reviewer 2:
Thank you very much for your comments. We have made the corrections you suggested highlighted in green. Details are in the attached file.
